LINC01936 inhibits the proliferation and metastasis of lung squamous cell carcinoma probably by EMT signaling and immune infiltration

Tian Qinqin 1 2
Liu Xiyao 2
Li Ang 3
Wu Hongjiao 3
Xie Yuning 3
Zhang Hongmei 3
Wu Fengjun 2
Chen Yating 2
Bai Congcong 2
Zhang Xuemei 2 3 jyxuemei@gmail.com
1 The Second Affiliated Hospital of Army Medical University, Department of Clinical Laboratory , Chongqing , China
2 North China University of Science and Technology, College of Life Science , Tangshan , China
3 North China University of Science and Technology, School of Public Health , Tangshan , China
Liu Jinhui
Electronic publication date: 2023 Dec 7
Publication date: 2023
Volume: 11
Electronic Location ID: e16447
Received 2023 Jun 30; Accepted 2023 Oct 21
Copyright: © 2023 Tian et al.
Copyright year: 2023
Copyright holder: Tian et al.
License: This is an open access article distributed under the terms of the Creative Commons Attribution License, which permits unrestricted use, distribution, reproduction and adaptation in any medium and for any purpose provided that it is properly attributed. For attribution, the original author(s), title, publication source (PeerJ) and either DOI or URL of the article must be cited.
License URL: https://creativecommons.org/licenses/by/4.0/

Keywords: LINC01936, lung squamous cell carcinoma, lncRNA, immune infiltration

Funding: Natural Science Foundation of Hebei province of China (No. H2017209233) This work was supported by the Key Project of the Natural Science Foundation of Hebei province of China (No. H2017209233). The funders had no role in study design, data collection and analysis, decision to publish, or preparation of the manuscript.

==============================
Purpose

To discover the biological function and potential mechanism of LINC01936 in the development of lung squamous cell carcinoma (LUSC).

Methods

Transcriptome data of LUSC from The Cancer Genome Atlas (TCGA) and Gene Expression Omnibus (GEO) databases were used to analyze the differentially expressed lncRNAs in LUSC and normal tissues by R “DEseq2”, “edgeR” and “limma” packages. The subcellular localization of LINC01936 was predicted by lncLocator. Cell proliferation and apoptosis were measured by CCK-8, MTT assay and Hoechst fluorescence staining. The migration and invasion were detected by Transwell assay. The function and pathway enrichment analysis were performed by Gene Ontology (GO) terms, Kyoto Encyclopedia of Genes and Genomes (KEGG) pathway analysis and gene set variation analysis (GSVA). The downstream targets of LINC01936 were predicted using RNA-Protein Interaction Prediction (RPISeq) program. The effect of LINC01936 on tumor immune infiltration was analyzed using Pearson Correlation Analysis using R “ggpubr” package.

Results

Based on the gene expression data of LUSC from TCGA database, 1,603, 1,702 and 529 upregulated and 536, 436 and 630 downregulated lncRNAs were obtained by DEseq2, edgeR and limma programs, respectively. For GSE88862 dataset, we acquired 341 differentially expressed lncRNAs (206 upregulated and 135 downregulated). Venn plot for the intersection of above differential expressed lncRNAs showed that there were 29 upregulated and 23 downregulated genes. LINC01936 was one of downregulated lncRNAs in LUSC tissues. The biological analysis showed that the overexpression of LINC01936 significantly reduced proliferation, migration and invasion of LUSC cells, and promoted cell apoptosis. The knockdown of LINC01936 promoted cell proliferation and metastasis. Pathway and GSVA analysis indicated that LINC01936 might participated in DNA repair, complement, cell adhesion and EMT, etc. LINC01936 was predicted to interact with TCF21, AOC3, RASL12, MEOX2 or HSPB7, which are involved in EMT and PI3K-AKT-MTOR pathway, etc. The expression of LINC01936 was also positively correlated with the infiltrating immune cells in LUSC.

Conclusions

LINC01936 is downregulated in LUSC. LINC01936 affected proliferation, migration and invasion of LUSC cells probably by EMT and immune infiltration, which might serve as a new target for the treatment of LUSC.

Introduction

Lung cancer is the most prevalent cancer worldwide and the leading cause of cancer-related mortality (Zhou et al., 2021). Lung squamous cell carcinoma (LUSC) is a common pathological type of lung cancer, which accounts for approximately 40–55% of lung cancer cases (Lu et al., 2021). LUSC cannot be easily diagnosed in its early stage, resulting in its poor prognosis (Miller et al., 2019). Therefore, there is an urge need for discovering novel and effective therapeutic targets for LUSC.

Long non-coding RNAs (lncRNAs) are a class of RNAs with more than 200 nucleotides in length and without protein-coding potential (Esteller, 2011). lncRNAs are primarily located in the nucleus, with a small amount present in the cytoplasm and organelles (Herbst, Morgensztern & Boshoff, 2018). Several studies have shown that many of the identified lncRNAs play a significant role as potential oncogenes or tumor suppressors in the development of various cancers (Peng, Koirala & Mo, 2017; Sun et al., 2020; Wu et al., 2020). Some lncRNAs had great impact on the development of LUSC by affecting immune response (Chi et al., 2019; Liu et al., 2020; Zhou et al., 2021). For example, lncRNA GNAS-AS1 promotes tumor progression in NSCLC by altering macrophage polarization (Li et al., 2020b). SNHG12 facilitated the immune escape of NSCLC by binding to HuR and increasing PD-L1 expression (Huang et al., 2022). lncRNA-ATB promotes the proliferation, migration and invasion of LUSC cells through the epithelial-mesenchymal transition (EMT), which is associated with immune escape in tumor microenvironment (Li et al., 2020a; Terry et al., 2017). The functions and mechanisms of immune-related lncRNAs in LUSC need to be further explored.

Long intergenic non-coding RNA 01936 (LINC01936, NR_122048.1), also known as NONHSAG027390.2 or HSALNG0013967, is located in chromosome 2p23.1. LINC01936 is mainly expressed in the lung, heart and thyroid tissues (Fagerberg et al., 2014). Hou & Yao (2021) reported that LINC01936 was downregulated and could serve as an independent prognostic gene in patients with lung adenocarcinoma. Chen, Ren & Cai (2021) constructed a lung adenocarcinoma survival-related lncRNA-miRNA-mRNA network and identified LINC01936 as one of the hub genes. In this study, after determining differential lncRNAs expression in LUSC, we analyzed the biological function and explored the potential mechanism of LINC01936 in LUSC cells. Our findings suggested that LINC01936 might serve as a promising target for treating LUSC. The workflow of this study is shown in Fig. 1.

Figure 1 Overview of the work flow.

Materials and Methods

RNA-seq data acquisition and preprocessing

The RNA-seq expression profiles of LUSC were downloaded from The Cancer Genome Atlas (TCGA, Version: V31.0). The TCGA-LUSC dataset contained an RNA expression profile from 502 LUSC tissues and 49 adjacent normal tissues. The lncRNA microarray dataset (GSE88862) was obtained from the Gene Expression Omnibus (GEO), which included three LUSC tissues and three adjacent normal tissues (Cheng et al., 2017).

Differential expression analysis

Differential expression analysis of lncRNAs between LUSC tissues and normal tissues were conducted using the DEseq2 (Love, Huber & Anders, 2014), edgeR (Robinson, McCarthy & Smyth, 2010) and limma (Ritchie et al., 2015) R packages. For the GEO dataset, the online analytical tool GEO2R was used to identify differentially expressed lncRNAs. The cut-off criteria were |log2FoldChange| > 2 and adj.p–value < 0.05. VENNY 2.1.0 was employed to draw Venn diagrams.

Differential expression analysis of LINC01936 in LUSC

The expression data of LINC01936 in LUSC tissues and normal tissues were extracted and processed by the Perl program (Version: V5.30.2). For the LINC01936 expression matrix from 502 LUSC tissues and 49 adjacent normal tissues, the beeswarm and limma packages were utilized to differentiate the expression level of LINC01936 between the tumor group and the normal group.

Prediction of subcellular localization of LINC01936

The lncLocator was used to predict the subcellular localization of lncRNA (Cao et al., 2018). The fasta sequence of LINC01936 (NR_122048.1) was downloaded and then uploaded to IncLocator. The five subcellular localizations (ribosomes, cytoplasm, cytosol, exosomes and nucleus) of LINC01936 were predicted.

Cell lines and culture conditions

LUSC cell lines (NCI-H1703, KNS-62) were purchased from Procell (Hubei, China). NCI-H1703 cells were cultured in RPMI 1640 medium (Gibco, Waltham, MA, USA) and KNS-62 was cultured in DMEM/High Glucose (Eallbio, Beijing, China) in a humidified incubator at 37 °C with 5% CO2. All medium supplemented with 10% Fetal Bovine Serum (FBS; Tianhang Biological, Zhejiang, China) and 1% Penicillin-Streptomycin (Solarbio, Beijing, China).

The synthesis and transfection of overexpression plasmid and siRNA

The full-length cDNA of LINC01936 was subcloned into the pcDNA3.1 vector to generate the pcDNA3.1/LINC01936 plasmids by Sangon Biotechnology (Shanghai, China) and verified by direct sequencing using the primer 5′- TGGGAGGTCTATATAAGCAGAG-3′(F). The empty pcDNA3.1 was used as negative control. The target sequence for LINC01936-specific small interfering RNA (siLINC01936; 5′-GGCAGACAUCCCACGGAAATT-3′) and control siRNA (siNC; 5′-UUCUCCGAACGUGUCACGUTT-3′) were synthesized by GenePharma (Shanghai, China). The vectors and siRNA were transfected into LUSC cells by using Lipofectamine 2000 transfection reagent (Invitrogen, Carlsbad, CA, USA).

RNA Extraction and Reverse Transcription-quantitative PCR (RT-qPCR)

Total RNA was extracted using TRIzol reagent (Invitrogen, Carlsbad, CA, USA) and 2 ug RNA was reverse transcribed into cDNA by using Revert Aid First Strand cDNA Synthesis Kit (Thermo Fisher Scientific, Grand Island, NY, USA) in accordance with the manufacturer’s instructions. LncRNA expression was measured using the ABIPRISM® 7900HT Fast Real-Time PCR system (Applied Biosystems, Foster City, CA, USA), and the reaction was performed using the Power SYBR Green PCR Master Mix (Thermo Fisher Scientific, Grand Island, USA). The cycling program was as follows: pre-denaturation at 95 °C for 2 min, followed by 45 cycles of denaturation at 95 °C for 15 s, annealing for 2 min, and final extension at 60 °C for 1 min. Three replicates per reaction were used in RT-qPCR, and the results were analyzed using the 2−ΔΔCt method. GAPDH was used as an internal reference. All primers for RT-qPCR amplification of candidate genes were listed in Table 1.

Table 1 Primer sequence for RT-qPCR.

Name of primer	Sequences	
LINC01936–F	5′-CCGCTGAATGGGGATTTTCG-3′	
LINC01936–R	5′-GTTTGGCAAAAACTGGGCTCT-3′	
GAPDH–F	5′-CTGGGCTACACTGAGGACC-3′	
GAPDH–R	5′-AAGTGGTCGTTGAGGGCAATG-3′	
Note:

F, forward; R, reverse.

Cell proliferation analysis

After transfected with LINC01936 overexpression plasmids or siRNA for 24 h, LUSC cells transfected for 24 h were seeded onto a 96-well plate at the density of 5 × 103 cells/well and cultured for 24, 48 or 72 h. For Cell Counting Kit-8 (CCK8) assay, 10 μL of CCK8 (Dojindo Molecular Technologies, Kumamoto, Japan) reagent was added to each well and the cells were incubated at 37 °C for 2 h. The optical density (OD) value was measured in an Infinite M200 PRO Microplate Reader (Tecan, Männedorf, Switzerland). For MTT assay, LUSC cells (2.5 × 103/well) was seeded on 96-well plate and incubated. After 24, 48 or 72 h, 100 μL MTT (Mackline, Shanghai, China) solution (0.5 mg/mL) was added to each well and incubated for 4 h. The supernatant was then sucked away and DMSO was added to melt the crystal. The OD490 was measured using a microplate reader. Each assay was performed in triplicate.

Apoptosis assay

The nuclear morphology of the apoptotic cells was visualized by Hoechst 33342 staining. After transfection with LINC01936 overexpressed plasmids for 24 h or si-LINC01936 for 48 h, NCI-H1703 cells were fixed with 4% paraformaldehyde (biosharp, Anhui, China) for 30 min and then stained with Hoechst 33342 solution (Solarbio, Beijing, China) for 20 min. The cells were observed under an IX71 inverted fluorescence microscope (Olympus, Tokyo, Japan). The apoptotic cells were quantified by Image J software (version 1.52a).

Migration and invasion assays

Transwell chambers (Corning Incorporated, Corning, NY, USA) with or without Matrigel were used for cell invasion and migration. LUSC cells (5 × 104) were added to the upper chambers in a total volume of 100 μL medium and 600 μL culture medium with 20% FBS was added to the lower chambers. LUSC cells were incubated for 24 h in migration assay and 48 h in invasion assay. The cells on upper chamber were swiped using cotton swabs and the cells on lower chamber were fixed with 4% paraformaldehyde for 30 min and stained with 0.1% crystal violet (Solarbio, Beijing, China) for 15 min. The migrated and invaded cells were counted using an IX71 inverted fluorescence microscope.

Functional and pathway enrichment analysis and targets prediction of lncRNA

Pearson correlation coefficient was used to screen the mRNAs co-expressed genes with LINC01936, with |r| > 0.65 and p < 0.05 as the thresholds criteria. The ggplot2 R package and the DAVID 6.8 database were used for Gene Ontology (GO) and Kyoto Encyclopedia of Genes and Genomes (KEGG) pathway enrichment analysis. Based on the median of the expression of LINC01936, LUSC samples were divided into high- and low-expression groups. The gene set variation analysis (GSVA) was carried out to study the pathways related to LINC01936 using GSVA R package. The gene set “2.cp.kegg.v6.2.symbols.gmt” from Molecular Signature Database served as the reference. The scores of LINC01936 binding to co-expression genes were calculated using RNA-Protein Interaction Prediction (RPISeq) (http://pridb.gdcb.iastate.edu/RPISeq/) (Muppirala, Honavar & Dobbs, 2011). Interaction probabilities generated by RPISeq. The probabilities (RF and SVM scores) >0.8 were considered as interacting gene.

Tumor-infiltrating immune analysis

Spearman correlation analysis was performed using ggpubr package in the R platform to analyze the correlation between the expression of LINC01936 and the abundance of immune cell (T cells, Natural Killer cells, Macrophages, Dendritic cells, CD8+ T cells and B cells) infiltration in LUSC. RNA-seq data was used for estimating the relative infiltration abundance of different immune cells.

Statistical analyses

Statistical analysis and visualization of results were conducted using SPSS (Version: 22.0; SPSS Inc., Chicago, IL, USA) and GraphPad Prism (Version: 8.0; GraphPad Software, La Jolla, CA, USA). The difference in the expression of LINC01936 between the tumor and normal groups was analyzed using the Wilcoxon rank-sum test. Differences of LINC01936 expression between the overexpressed/silencing group and the control group were analyzed using Student’s t-test. p < 0.05 was considered statistically significant.

Results

Identification of differentially expressed lncRNAs in LUSC

By using DESeq2, edgeR, and limma packages, 2,139 (1,603 upregulated and 536 downregulated), 2,138 (1,702 upregulated and 436 downregulated), and 1,159 (529 upregulated and 630 downregulated) differentially expressed lncRNAs were identified in the TCGA-LUSC dataset, respectively (Figs. 2A–2C). For GEO GSE88862 dataset, 341 differentially expressed lncRNAs (206 upregulated and 135 downregulated) were identified (Fig. 2D). The intersection analysis of the two datasets revealed 52 co-differentially expressed lncRNAs, with 29 upregulated and 23 downregulated (Figs. 2E and 2F). Among these differentially expressed genes, the top four down-regulated lncRNAs were SMIM25, FENDRR, SFTA1P, and LINC01936 (Table 2) and the top four up-regulated ones were LINC01133, CASC9, SOX21-AS1, and ESRG (Table 3). After searching the literatures, we found that seven genes, except for LINC01936, were all contributed to the development of lung cancer by several pathways (Chang et al., 2020; Gao et al., 2019; Jafari et al., 2022; Jiang et al., 2020; Ning et al., 2021; Wang et al., 2021a; Xiong et al., 2019). Due to this, LINC01936 was used for further analysis.

Figure 2 Differentially expressed lncRNAs in LUSC tissues and normal tissues.

Volcano plots show the differentially expressed lncRNAs using DEseq2 (A), edgeR (B) and limma (C) packages in the TCGA database. (D) The volcano plot shows the distribution of the differentially expressed lncRNAs in GSE88862. The Venn plot shows common differentially upregulated lncRNAs (E) and differentially downregulated lncRNAs (F) in the TCGA database and the GSE88862 dataset.

Table 2 The Expression level of the 23 downregulated lncRNAs.

Gene	Expression in tumor tissues	Expression in adjacent tissues	Log2FC	adj.p value	
SMIM25	382.17	3,602.77	−3.62	1.04E–85	
FENDRR	204.94	3,809.76	−4.98	2.13E–106	
SFTA1P	156.93	4,278.69	−6.04	3.93E–119	
LINC01936	75.96	917.06	−4.15	6.12E–82	
LHFPL3–AS2	75.83	1,646.84	−5.72	1.73E–92	
TARID	31.41	208.00	−3.89	1.11E–66	
MYO16–AS1	14.39	137.02	−4.35	6.89E–64	
LINC01612	9.71	69.69	−4.32	2.92E–64	
LINC01290	8.88	56.96	−2.64	8.13E–64	
LINC01765	6.54	68.14	−4.14	7.64E–51	
LINC01996	5.78	203.02	−6.57	3.84E–153	
LINC01513	3.62	23.39	−2.81	1.22E–28	
LINC00165	3.38	23.18	−3.35	1.33E–32	
LINC01166	2.87	21.10	−3.36	1.65E-40	
LINC01827	2.66	18.55	−3.36	2.89E–45	
LINC01616	2.27	12.76	−3.10	5.89E–40	
LINC01863	2.09	49.88	−5.16	3.57E–119	
LINC02016	2.00	183.24	−6.48	2.82E–147	
PGM5–AS1	1.22	21.71	−4.20	8.79E–90	
SLC14A2–AS1	1.19	9.49	−2.82	3.08E–42	
LINC01412	1.11	9.96	−2.79	1.82E–40	
LINC01985	0.84	11.00	−3.32	1.28E–59	
LINC00844	0.74	12.41	−3.57	7.56E–67	

Table 3 The Expression level of the 29 upregulated lncRNAs

Gene	Expression in tumor tissues	Expression in adjacent tissues	Log2FC	adj.p value	
LINC01133	62.04	1049.26	3.20	1.42E-08	
CASC9	5.08	743.48	7.74	3.05E-33	
SOX21-AS1	39.65	597.57	3.51	1.18E-13	
ESRG	2.57	359.98	2.37	1.36E-04	
SH3PXD2A-AS1	21.98	264.94	2.46	8.36E-10	
LINC01116	39.53	242.06	2.19	2.44E-10	
LINC00626	0.61	137.22	4.71	1.16E-26	
SLC2A1-AS1	14.33	130.70	3.09	2.48E-37	
LINC01564	6.55	105.67	3.55	3.96E-14	
PCAT7	6.84	100.61	3.47	6.25E-24	
LINC00640	7.29	70.06	2.75	4.53E-16	
LINC01305	0.51	69.85	4.53	2.49E-35	
LINC02041	11.96	68.83	2.05	1.80E-09	
LINC01752	3.98	64.78	3.30	4.43E-15	
LINC01807	0.31	56.80	4.76	1.28E-30	
ABCA9-AS1	2.39	48.24	3.68	1.63E-17	
LINC01977	2.53	34.60	3.33	1.91E-30	
LINC02163	0.08	33.71	5.02	4.56E-59	
LINC02178	0.24	32.54	2.67	4.54E-20	
LINC01705	2.20	27.43	3.15	8.57E-21	
LUARIS	1.14	25.53	3.38	4.22E-29	
MAFA-AS1	0.78	25.11	3.20	2.73E-25	
LSAMP-AS1	0.92	24.50	3.44	8.14E-26	
LINC01117	3.92	19.61	2.03	2.07E-10	
LINC01905	1.43	17.84	2.87	5.67E-19	
LINC01633	0.02	16.78	4.28	1.10E-70	
FAM83C-AS1	1.16	7.33	2.26	1.77E-30	
LINC02473	0.16	7.10	2.41	2.33E-27	
KCNQ5-IT1	0.14	4.68	2.08	1.81E-23	

The expression and localization of LINC01936 in LUSC

The expression of LINC01936 in LUSC tissues was significantly downregulated in LUSC tissues as compared to that in normal tissues (p < 0.001, Fig. 3A). In addition, the subcellular localization of LINC01936 was predicted using lncLocator, and result showed that LINC01936 was mainly localized in the cytoplasm and organelles with a total score of 0.89 (Fig. 3B).

Figure 3 The expression and localization of LINC01936 in LUSC.

(A) The expression level of LINC01936 in LUSC tissues and normal tissues. (B) Results of the cytoplasmic-nuclear localization prediction of LINC01936. (C) The overexpression efficiency of LINC01936 in LUSC cells was studied by RT-qPCR. (D) The knockdown efficiency of LINC01936 in LUSC cells was studied by RT-qPCR. *p < 0.05, **p < 0.01, ***p < 0.001.

The effect of LINC01936 on the proliferation and metastasis of LUSC cells

To study the biological function of LINC01936 in LUSC cells, we measured the RNA expression in LINC01936 was overexpressed or knockdown by transfecting with pcDNA3.1/LINC01936 plasmid and siLINC01936, respectively. LINC01936 and siLINC01936 were successfully transfected into LUSC cells (Figs. 3C and 3D). The effect of LINC01936 on cell proliferation and viability was evaluated by performing CCK-8 assay and MTT assay. The results indicated that LINC01936 overexpression reduced cell proliferation and viability (Figs. 4A–4D) and the silence of LINC01936 promoted cell proliferation and viability (Figs. 5A–5D). Transwell assays were conducted to investigate the effects of LINC01936 on cell migration and invasion. The results showed that the overexpression of LINC01936 significantly reduced the migration and invasion abilities of LUSC cells (Figs. 6A and 6B). In contrast, the silence of LINC01936 enhance the cell migration and invasion (Figs. 6C and 6D). Taken together, these results suggest that LINC01936 suppresses the proliferative and metastatic abilities of LUSC cells.

Figure 4 The effects of overexpression of LINC01936 on cell proliferation and cell viability.

The effect of overexpression LINC01936 on the proliferation of NCI-H1703 (A) and KNS-62 (B) cells was detected by CCK-8 assay. The effect of overexpression of LINC01936 on the viability of NCI-H1703 (C) and KNS-62 (D) cells was detected by MTT assay. ns, p > 0.05, *p < 0.05, **p < 0.01, ***p < 0.001.

Figure 5 The silencing of LINC01936 promoted the proliferation and viability of LUSC cells.

The effect of silencing LINC01936 on the proliferation of NCI-H1703 (A) and KNS-62 (B) cells was detected by CCK-8 assay. The effect of silencing LINC01936 on the viability of NCI-H1703 (C) and KNS-62 (D) cells was detected by MTT assay. *p < 0.05, **p < 0.01, ***p < 0.001.

Figure 6 Effects of LINC01936 abnormal expression on LUSC cell migration and invasion.

The migration and invasion of NCI-H1703 (A) and KNS-62 (B) cells in overexpression LINC01936 groups were determined by Transwell assay (magnification, ×100). The migration and invasion of NCI-H1703 (C) and KNS-62 (D)cells in siLINC01936 groups were determined by Transwell assay (magnification, ×100).

LINC01936 promotes the apoptosis of LUSC

To investigate the effect of LINC01936 on cell apoptosis, nuclear fluorescence staining was performed using Hoechst33342. Results showed that the number of bright blue regions, which indicating condensed DNA and fragmented nuclei, was significantly higher in pcDNA3.1/LINC01936 transfected group than in control group (p < 0.01, Figs. 7A and 7B) and the silence of LINC01936 inhibited cell apoptosis (Figs. 7C and 7D). These findings confirmed that LINC01936 enhanced the apoptosis of LUSC cells.

Figure 7 The effects of LINC01936 on apoptosis in LUSC.

Hoechst 33342 fluorescent staining was used to detect apoptosis in NCI-H1703 (A) and KNS-62 (B) cells after overexpression of LINC01936, as well as apoptosis of NCI-H1703 (C) and KNS-62 (D) cells after silencing LINC01936.

Functional enrichment and targets of LINC01936 analysis

GO analysis revealed significant enrichment of LINC01936 in biological processes related to cell matrix adhesion, regulation of endothelial cell proliferation and regulation of angiogenesis (Fig. 8A). The cell components were mainly concentrated in focal adhesion, bicellular tight junction and collagen trimer (Fig. 8B). Molecular functions were significantly enriched in extracellular matrix structural constituent, integrin binding and growth factor binding (Fig. 8C). KEGG pathway enrichment analysis indicated that LINC01936 was mainly involved in cell adhesion molecules, vascular smooth muscle contraction, cGMP−PKG signaling pathway and leukocyte transendothelial migration (Fig. 8D). To further investigate the potential mechanism of LINC01936 in LUSC, Gene Set Variation Analysis (GSVA) was performed on the TCGA-LUSC RNA-seq data (Fig. 8E). The high expression of LINC01936 significantly deactivation the epithelial mesenchymal transition (EMT).

Figure 8 Functional and pathway enrichment analysis.

The top 10 terms in Biological Process (A), Cell Components (B) and Molecular Function (C) in GO enrichment analysis. (D) The KEGG pathways (top 10) in which the co-expressed mRNAs were enriched. (E) GSVA-derived clustering heatmaps of differentially expressed genes in the LINC01936 high expression group.

Next, the downstream targets of LINC01936 were predicted. Prior studies suggested that lncRNAs in the cytoplasm can bind to cytoplasmic RNAs to affect the expression patterns of downstream targets (Montes et al., 2015; Suravajhala et al., 2015). Correlation analysis showed that LINC01936 was correlated with the expression of transcription factor 21 (TCF21) (r = 0.758, p < 0.001), sodium voltage-gated channel alpha subunit 7 (SCN7A) (r = 0.752, p < 0.001), amine oxidase copper containing 3 (AOC3) (r = 0.738, p < 0.001), C1q and TNF related 7 (C1QTNF7) (r = 0.734, p < 0.001), Cas scaffold protein family member 4 (CASS4) (r = 0.711, p < 0.001), phospholipase A2 group V (PLA2G5) (r = 0.697, p < 0.001), microfibril associated protein 4 (MFAP4) (r = 0.696, p < 0.001), gap junction protein alpha 5 (GJA5) (r = 0.684, p < 0.001), RAS like family 12 (RASL12) (r = 0.678, p < 0.001), mesenchyme homeobox 2 (MEOX2) (r = 0.677, p < 0.001), indolethylamine N-methyltransferase (INMT) (r = 0.675, p < 0.001), T-box transcription factor 5 (TBX5) (r = 0.668, p < 0.001), Heat shock protein B7 (HSPB7) (r = 0.666, p < 0.001), and podocan (PODN) (r = 0.659, p < 0.001) in LUSC tissues (p < 0.001, Fig. 9A). Further interaction analysis presented that LINC01936 might bind to TCF21, AOC3, RASL12, MEOX2 and HSPB7 with RF and SVM scores greater than 0.8 (Figs. 9B and 9C).

Figure 9 Prediction of downstream target of LINC01936.

(A) Correlation analysis between LINC01936 and co-expressed genes. (B) and (C) The binding probability of LINC01936 and co-expressed genes was predicted by RPISeq database. p > 0.05, *p < 0.05, **p < 0.01, ***p < 0.001.

LINC01936 is correlated with immune cells infiltrating

To further identify the specific cell types that played a major role in the process of LUSC, the potential association between LINC01936 expression and immune cell infiltration was analyzed based on TCGA-LUSC transcriptomic data. The results showed that LINC01936 was highly expressed in all infiltrating immune cells (Fig. 10A) and the expression of LINC01936 was significantly correlated with T cells (r = 0.30, p < 0.05), Natural Killer cells (r = 0.11, p < 0.05), Macrophages (r = 0.45, p < 0.05), Dendritic cells (r = 0.34, p < 0.05), CD8+ T cells (r = 0.24, p < 0.05) and B cells (r = 0.30, p < 0.05) in LUSC (Fig. 10B). LINC01936 expression was also significantly correlated with the proportion of infiltrating immune cells, which may affect the occurrence and development of LUSC.

Figure 10 The correlation of LINC01936 expression and immune cells in LUSC.

(A) The relationship between the LINC01936 expression level and immune cells. (B) The correlation between LINC01936 expression and immune cells.

Discussion

LUSC is a malignant tumor with multiple etiologies. The pathogenetic mechanisms remain unclear. lncRNA is a sequence-conserved RNA with unstable expression in several species. The diagnostic sensitivity of lncRNA is significantly higher than that of DNA and mRNA (Krepischi, Pearson & Rosenberg, 2012). In present study, we discovered the biological function of LINC01936 in LUSC, conducted pathway enrichment and immune cell infiltration analysis.

Following the development of microarray and high-throughput sequencing technologies, studies have shown that dysregulated of key lncRNAs influence the process of various cancers (Zhang et al., 2019). DEseq2 and edgeR programs are suitable only for the differential expression analysis of the sequencing data, and limma package was for both microarray and sequencing data (Love, Huber & Anders, 2014; Ritchie et al., 2015; Robinson, McCarthy & Smyth, 2010). We obtained differentially expressed lncRNAs using RNA-seq data from TCGA and GEO database by using these three programs. After co-differential expression analysis, we obtained 52 co-differentially expressed lncRNAs (29 upregulated and 23 downregulated). The top-four upregulated ones were LINC01133, CASC9, SOX21-AS1, and ESRG. Studies presented that LINC01133, CASC9 and SOX21-AS1were highly expression in lung cancer tissues and its knockdown could inhibited cell proliferation or invasion (Gao et al., 2019; Lu et al., 2017; Zhang, Zhu & Chen, 2015). Based on present study, the top-four downregulated lncRNAs were SMIM25, FENDRR, SFTA1P, and LINC01936. This is consistent with previous studies, which demonstrated that SMIM25 and FENDRR were significantly downregulated in lung cancer (Pan et al., 2020; Tang et al., 2021).

Currently, the bioinformatics analysis showed that LINC01936 was downregulated in lung adenocarcinoma tissues and contributed to a good prognosis of lung adenocarcinoma patients (Hou & Yao, 2021). In this study, we demonstrated that LINC01936 was downregulated in LUSC tissues.

A ceRNA network analysis showed that LINC01936 mediated the activity of miR-20a-5p to regulate TGF-β signaling and downstream pathways to affect the progression and prognosis of lung adenocarcinoma (Chen, Ren & Cai, 2021). In present study, we found that LINC01936 could inhibit the proliferation, migration and invasion of LUSC cells and promote cell apoptosis; which indicated that LINC01936 might serve as a tumor suppressor gene to affect LUSC development. The growth and metastases of LUSC cells were closely related to lncRNAs in LUSC; for example, lncRNA-ATB (Li et al., 2020a) and LINC01272 (Ma et al., 2021).

To further investigate the potential mechanism of LINC01936 in LUSC, GO and KEGG enrichment analyses and GSVA were performed. GSVA presented that the high expression of LINC01936 was associated with the deactivation of EMT and angiogenesis. The EMT process is a complex phenotypic event that prompts cells to abandon their extensive epithelial cell-cell contacts, apical basal polarity and distinct cytoskeletal architecture to become more motile and invasive (Lamouille, Xu & Derynck, 2014). High occurrences of metastasis are usually associated with EMT in most malignant epithelial tumors (Wang et al., 2021b). Notably, lncRNAs are described as an important regulatory RNA molecule to influence the EMT process at multiple levels (Grelet et al., 2017). In the experiment part, we found that LINC01936 involved in the process of cell invasion and migration. Meanwhile, GSVA also confirmed that the high expression of LINC01936 was associated with the deactivation of EMT. The GSVA results also showed that the high expression of LINC01936 was significantly related to the complement, angiogenesis, IL6-JAK-STAT3 signaling pathway, and PI3K-AKT-MTOR pathway. These findings provided new insights into the pathogenesis of LUSC.

It is worth noting that RPISeq database prediction indicated a comparatively high probability of LINC01936 binding to transcription factor 21 (TCF21), amine oxidase copper containing 3 (AOC3), RAS like family 12 (RASL12), mesenchyme homeobox 2 (MEOX2) and heat shock protein B7 (HSPB7). These proteins mainly involved in cell invasion by controlling the EMT biological processes. For instance, the upregulation of TCF21 inhibits the metastasis of esophageal squamous cell carcinoma and colorectal cancer (Chen et al., 2018; Dai et al., 2016; Dai et al., 2017). AOC3 is an endothelial adhesion protein, and low-level AOC3 facilitated mesenchymal transformation and decreased CD4+ T cell recruitment to lung cancer (Chang et al., 2021). The decreased AOC3 levels are correlated with lymph node and hepatic metastasis in colorectal cancer (Toiyama et al., 2009). AOC3 is an endothelial adhesion protein, the low-level AOC3 facilitated mesenchymal transformation (Chang et al., 2021) and correlated with lymph node and hepatic metastasis in colorectal cancer (Toiyama et al., 2009). MEOX2 inhibits cell proliferation and EMT in vascular smooth muscle and endothelial cells (Gorski & Leal, 2003; Valcourt et al., 2007). A study also showed that HSPB7 was downregulated in endometrial carcinoma (EC) and influenced EC cell proliferation and metastasis via the PI3K/AKT/mTOR signaling pathway (Xing, Wu & Wang, 2023). LINC01936 may mainly inhibit the development of LUSC by deactivating EMT. Studies have shown that ‘mesenchymal’ features of LUSC are associated with exclusion of activated T-cells in the tumor microenvironment (TME), which in turn promotes the incretion of inflammatory cytokines and upregulation of immunosuppressive immune checkpoint factors (Chae et al., 2018).

In recent years, studies have shown that the interaction between tumor cells and tumor microenvironment plays a key role in the process of tumor occurrence and development and has an important impact on the efficacy of immunotherapy (Di Modugno et al., 2019; Fu et al., 2019). The immunosuppressive microenvironment induced by T lymphocytes weakens the antitumor immune function of cells and is a key factor in antitumor immunotherapy (Bense et al., 2016; Campa et al., 2016; Kurebayashi et al., 2016). Previous studies have shown that the level of immune cell infiltration in LUSC was higher than that in other histopathological which suggested the important function of immune environment in LUSC (Li et al., 2016; Varn et al., 2017). This study analyzed the relationship between the expression level of LINC01936 and the infiltration level of immune cells in LUSC. We demonstrated that the expression of LINC01936 was positively correlated with the infiltration of T cells, natural killer cells, macrophages, dendritic cells, CD8+ T cells and B cells. Our study suggests that LINC01936 may affect the development of LUSC through immune pathways.

Conclusions

LINC01936 attenuated cell proliferation, migration and invasion and promoted cell apoptosis in LUSC probably by EMT and immune infiltration. LINC01936 might be served as a therapeutic target for LUSC.

Supplemental Information

Supplemental Information 1 Differentially expressed lncRNAs in LUSC tissues and normal tissues in The Cancer Genome Atlas.

The differentially expressed lncRNAs from LUSC and normal tissues were collected using the DEseq2, edgeR and Limma packages in R platform.

Click here for additional data file.

Supplemental Information 2 Differentially expressed lncRNAs in LUSC tissues and normal tissues in GSE88862 dataset.

The differentially expressed lncRNAs from GSE88862 was identified by the online analytic tool GEO2R provided by the GEO database.

Click here for additional data file.

Supplemental Information 3 Differentially expressed lncRNAs in LUSC tissues and normal tissues in TCGA and GSE88862 datasets.

VENNY 2.1.0 was employed to draw Venn diagrams based on the differentially expressed lncRNAs.

Click here for additional data file.

Supplemental Information 4 The expression level of LINC01936 in lung squamous cell carcinoma.

The beeswarm and Limma packages were utilized to differ the expression level of LINC01936 in tumor group from normal group.

Click here for additional data file.

Supplemental Information 5 Functional and pathways enrichment analysis.

The ggplot2 R package and the DAVID 6.8 database were used to establish Gene Ontology (GO) and Kyoto Encyclopedia of Genes and Genomes (KEGG) analysis. The GSVA R package was used to study the pathways related to LINC01936.

Click here for additional data file.

Supplemental Information 6 The downstream targets of LINC01936 analysis.

The scores of LINC01936 binding to co-expression genes were calculated using RNA-Protein Interaction Prediction (RPISeq). The probabilities (RF and SVM scores) >0.8 were considered as interacting gene.

Click here for additional data file.

Supplemental Information 7 The correlation of LINC01936 expression and immune cells in LUSC.

Pearson Correlation Analysis with ggpubr R package was used for analysis of tumor-infiltrating immune cells, and the correlations between the infiltrating level of immune cells.

Click here for additional data file.

Supplemental Information 8 The overexpression/knockdown efficiency of LINC01936 in LUSC cells was studied by RT-qPCR.

LINC01936 expression level was measured using the ABIPRISM® 7900HT Fast Real-Time PCR and reaction condition was conducted utilizing the Power SYBR Green PCR Master Mix.

Click here for additional data file.

Supplemental Information 9 The effect of LINC01936 on LUSC cells viability was detected by MTT.

The OD value of transfected cells was measured at 490 nm using the Infinite M200 PRO Microplate Reader.

Click here for additional data file.

Supplemental Information 10 The effect of LINC01936 on LUSC cells proliferation was detected by CCK-8.

The OD value of transfected cells was measured at 450 nm using the Infinite M200 PRO Microplate Reader, and the proliferation curve was drawn.

Click here for additional data file.

Supplemental Information 11 The migration of LUSC cells were determined by Transwell.

Migrated cells after overexpression of LINC01936 were photographed under IX71 inverted fluorescence microscope (magnification, ×100)

Click here for additional data file.

Supplemental Information 12 The migration of LUSC cells were determined by Transwell.

Migrated cells after silencing LINC01936 were photographed under IX71 inverted fluorescence microscope (magnification, ×100)

Click here for additional data file.

Supplemental Information 13 The invasion of LUSC cells were determined by Transwell.

Cells invaded after overexpression of LINC01936 were photographed under IX71 inverted fluorescence microscope (magnification, ×100)

Click here for additional data file.

Supplemental Information 14 The invasion of LUSC cells were determined by Transwell.

The migrated cells after silencing LINC01936 were photographed under IX71 inverted fluorescence microscope (magnification, ×200)

Click here for additional data file.

Supplemental Information 15 The apoptotic cells were detected by Hoechst33342 fluorescence stain.

Hoechst33342 fluorescence staining was used to detect the effect of overexpression of LINC01936 in H1703 on apoptosis.

Click here for additional data file.

Supplemental Information 16 The apoptotic cells were detected by Hoechst33342 fluorescence stain.

Hoechst33342 fluorescence staining was used to detect the effect of overexpression of LINC01936 in KNS-62 on apoptosis.

Click here for additional data file.

Supplemental Information 17 The apoptotic cells were detected by Hoechst33342 fluorescence stain.

Hoechst33342 fluorescence staining was used to detect the effect of silencing LINC01936 in H1703 on apoptosis.

Click here for additional data file.

Supplemental Information 18 The apoptotic cells were detected by Hoechst33342 fluorescence stain.

Hoechst33342 fluorescence staining was used to detect the effect of silencing LINC01936 in KNS-62 on apoptosis.

Click here for additional data file.

Studies were carried out in the North China University of Science and Technology, College of Life Science.

Additional Information and Declarations

Competing Interests

Author Contributions

Data Availability

The authors declare that they have no competing interests.

Qinqin Tian conceived and designed the experiments, performed the experiments, analyzed the data, prepared figures and/or tables, authored or reviewed drafts of the article, and approved the final draft.

Xiyao Liu performed the experiments, analyzed the data, prepared figures and/or tables, and approved the final draft.

Ang Li conceived and designed the experiments, prepared figures and/or tables, and approved the final draft.

Hongjiao Wu analyzed the data, prepared figures and/or tables, and approved the final draft.

Yuning Xie analyzed the data, prepared figures and/or tables, and approved the final draft.

Hongmei Zhang analyzed the data, prepared figures and/or tables, and approved the final draft.

Fengjun Wu performed the experiments, prepared figures and/or tables, and approved the final draft.

Yating Chen performed the experiments, prepared figures and/or tables, and approved the final draft.

Congcong Bai performed the experiments, prepared figures and/or tables, and approved the final draft.

Xuemei Zhang conceived and designed the experiments, authored or reviewed drafts of the article, and approved the final draft.

The following information was supplied regarding data availability:

The raw measurements are available in the Supplemental Files.

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
