# Peer review of "LINC01936 inhibits the proliferation and metastasis of lung squamous cell carcinoma probably by EMT signaling and immune infiltration"

_PeerJ, doi:10.7717/peerj.16447_

## Round 0.1 · original submission · Major Revisions

Authors should revise according to the suggestions of reviewers. The modifications should be marked. A point to point response letter is needed.

**Language Note:** PeerJ staff have identified that the English language needs to be improved. When you prepare your next revision, please either (i) have a colleague who is proficient in English and familiar with the subject matter review your manuscript, or (ii) contact a professional editing service to review your manuscript. PeerJ can provide language editing services - you can contact us at [email protected] for pricing (be sure to provide your manuscript number and title). – PeerJ Staff

Reviewer 1 ·

Basic reporting

no comment

Experimental design

The cell line used in the experiment was not enough.

Validity of the findings

no comment

Additional comments

In this manuscript entitled ‘LINC01936 mediates the progression of lung squamous cell carcinoma by reducing tumor cell proliferation and metastasis’, authors scrutinized the function of LINC01936 on the biological behavior of lung squamous cell cancers and discussed the possible molecular mechanism. There were some problems in this manuscript. Please see my comments below:

1.The cell line used in the experiment was not enough.
2.The work currently has language deficiencies that make the paper difficult to read. I recommend a linguistic revision of the work. Besides, there are some mistakes. For example, in Line 170, “d2ifferentially” should be corrected as “differentially”; in Line 297 “Studies was carried” need to be corrected as “Studies were carried”.
3.Some expressions are not rigorous. In Line 179, “LINC01936 was ranked fourth among the 23 downregulated lncRNAs in adjacent normal tissues”, I think the author mistakenly wrote “LUSC tissues” as “adjacent normal tissues”.
4.In Line 257-258, the statement “We found that LINC01936 could inhibit the proliferation, migration, and invasion of LUSC cells and promote their apoptosis; this finding indicated that LINC01936 functioned as a tumor suppressor gene to regulate LUSC development…” contradicts that the sentence in Line 275, “LINC01936 may participate in LUSC metastasis through the EMT and angiogenesis pathways…”.
5.In Line 249-250, “LINC01936 is downregulated…can thus serve as an independent prognostic factor for lung adenocarcinoma” is misleading for there is no evidence supporting this standpoint.
6.A flowchart of the work is suggested for better understanding.

Reviewer 2 ·

Basic reporting

The manuscripts well written with proper references in the introduction and discussion section. The codes provided are easily accessible and will be helpful in reproducing the results from this study.

Experimental design

The authors describe in their results that LINC01936 is ranked fourth among the 23 down regulated lncRNAs. They should also mention the other top 3 lncRNAs they identified in their analysis and the basis for only studying this particular lncRNA. Also, what were the top 3 up-regulated lncRNA identified in the analysis? The reason behind going ahead with only analyzing this particular lncRNA is not clear from the manuscript and should be included in the result/discussion section.

Validity of the findings

No comment

Reviewer 3 ·

Basic reporting

In this manuscript, the authors have investigated the role of LINC01936 in the progression of LUSC carcinoma using bioinformatics tools. The study revealed that LUSC tumor tissues exhibit reduced expression of LINC01936 and its expression is potentially positively correlated with the recruitment of various immune cells. This research contributes valuable insights to the field of lung cancer. However, the manuscript requires some revisions to enhance its clarity and support its claims.

Major Revisions:

1) In Fig 3C, the authors should elaborate on their approach to accurately quantify apoptotic cells, as manual counting may lead to misleading results. To strengthen their findings, the authors are encouraged to perform additional assays such as cell titer glo/MTT assay, which offers more quantifiable data.

2) To bolster their claims, the authors should consider silencing LINC01936 and quantifying the impact on LUSC proliferation or multiplication. This experimental approach will provide additional evidence supporting the role of LINC01936 in LUSC progression.

3)It is advisable for the authors to use bioinformatics predictions to identify and discuss the downstream targets of LINC01936. This analysis will help determine whether the effects of LINC01936 in LUSC are primarily cell autonomous or non-autonomous.

Minor Revisions:

1) The entire manuscript should be thoroughly proofread to correct grammatical errors and improve sentence formation. Specifically, the last paragraph of the discussion section requires revision for clarity and coherence.

2) The authors should address any repetitive titles present throughout the manuscript.

3) The main title of the manuscript needs to be revised to better reflect the content and findings of the study.

Experimental design

No comment

Validity of the findings

No comment

---

## Round 0.2 · accepted · Accept

The authors have addressed the reviewers' concerns properly and revised the manuscript accordingly. The manuscript can be accepted for publication in its current form.

Note the suggestion from Reviewer 2 regarding the article title.

Reviewer 1 ·

Basic reporting

no comment

Experimental design

no comment

Validity of the findings

no comment

Additional comments

no comment

Reviewer 2 ·

Basic reporting

The authors have addressed the queries raised before and improved the manuscript. I do not however recommend using the word "probably" in the title. Could the authors replace it with other synonymous word (e.g plausibly?).

Experimental design

no comment

Validity of the findings

no comment

Reviewer 3 ·

Basic reporting

NA

Experimental design

NA

Validity of the findings

NA